# The Effect of Domain-Specific Sitting Time and Exercise Habits on Metabolic Syndrome in Japanese Workers: A Cross-Sectional Study

**DOI:** 10.3390/ijerph17113883

**Published:** 2020-05-30

**Authors:** Rina So, Tomoaki Matsuo

**Affiliations:** 1Occupational Epidemiology Research Group, National Institute of Occupational Safety and Health, Kawasaki 214-8585, Japan; matsuo@crf-res.net; 2Research Center for Overwork-Related Disorders, National Institute of Occupational Safety and Health, Kawasaki 214-8585, Japan

**Keywords:** sedentary behavior, occupational health, leisure activity, sitting, physical activity

## Abstract

The effects of domain-specific (i.e., occupational, leisure-time on workday, and holiday) sitting time (ST), and exercise on metabolic syndrome (MetS) development are insufficiently studied. The present study aimed to examine the single and combined effects of each domain-specific ST and exercise habits on MetS. The total and domain-specific STs of 5530 participants were collected using a validated questionnaire. The multiple logistic regression analyses determined the effects of each domain-specific ST and exercise habit on MetS. Of all participants, 7.8% had MetS. Odds ratios (ORs) for MetS were significant only in the group with the longest leisure-time ST on holidays (OR, 1.43; 95% confidence interval [CI], 1.12–1.83); we found no significant associations with any other domain-specific ST after statistical adjustment for confounders. The no-habitual-exercise group clearly had a higher risk for MetS (OR, 1.44; 95% CI, 1.15–1.80). The significantly higher ORs for MetS was shown in only the combined longer total ST (OR, 1.64; 95% CI, 1.12–2.39) and holiday ST (OR, 1.83; 95% CI, 1.30–2.59) with no habitual exercise. These findings suggested that accumulated daily total ST, particularly leisure-time ST on holidays with no-habitual exercise, can increase the risk of MetS and it could possibly be mitigated by habitual exercise.

## 1. Background

Metabolic syndrome (MetS) is a large and growing public health problem [1,2]. Because of the health consequences associated with the high prevalence of MetS, understanding the determinants of MetS is important. In recent years, sedentary behavior (SB) has been identified as a significant public health issue [3]. Partly because of the automation of manual tasks through technological advancement, many adults spend one third to one half of their working hours [4,5,6] as well as a significant amount of leisure time [7,8] in SB. A recent meta-analysis by Edwardson et al. [9] found that those who spent large amounts of time sitting increased their odds of developing MetS by 73%. However, most of these studies focused on fragments of daily activity, such as watching television (TV) or using computers in leisure time. Nevertheless, sitting time (ST) occurs generally across numerous domains on a daily basis [10]. To fully understand the ST role as a factor related to health risks, it is necessary to identify whether various unequal domain-specific ST (i.e., occupational and leisure-time ST on workdays and holidays) have a similar effect on MetS. In addition, there is evidence which suggests that the adverse health effects of different domain ST combined may be greater than those associated with each ST separately [11]. Therefore, it is important to consider the unequal contribution of each specific-domain ST and the total amount of ST.

A high level of physical activity (PA) is inversely associated with the prevalence of MetS, and is well recognized as beneficial in preventing and managing MetS risk [12,13]. Previous studies [14,15,16] have attempted to clarify the relationship between ST, PA, and MetS, and have found that meeting PA recommendations (equivalent to 150 min/week of moderate to vigorous PA) may mitigate the detrimental association between SB and MetS. However, to date, studies examining the effect of both ST and PA on MetS have been limited by their application to a binary definition of MetS [16,17], the use of nonrepresentative samples (e.g., older adults [18]), or a focus on fragments of daily activity such as leisure time [16,19]. In addition, these studies were restricted to a categorical or quantitative evaluation of PA, with no findings on habitual exercise in terms of exercise frequency and duration. Thus, independent and combined analysis of domain-specific ST and habitual exercise is needed, which would help to target populations with MetS and to develop preventive guidelines.

To our knowledge, the effects of various domain-specific ST and habitual exercise on MetS have not been fully evaluated. Therefore, this study aimed, first, to examine the single effect of domain-specific ST and habitual exercise on MetS risk, and second, to clarify the combined associations of domain-specific ST and habitual exercise with respect to MetS in working adults.

## 2. Methods

### 2.1. Participants and Procedures

This cross-sectional, internet-based survey was conducted from January to July 2018 with participants answering a self-completed questionnaire. Participants were recruited with the goal of sampling a wide range of employment types, based on the composition ratio of employed persons according to sex, age group (20 to 65 years old), and industry type listed in the 2017 Japan Labor Force Survey (Ministry of Internal Affairs and Communications) [20]. The invitation email containing the details of the survey was sent to enrolled workers at an internet research company who met the basic conditions (sex, age, etc.). If they agreed to participate, the internet survey was administered, and they earned internet use points according to their answers. To ensure the reliability of the responses, we asked participants to prepare their medical checkup information from within 1 year before start of this survey, and when they were not prepared, they could not proceed to the main survey. Through the internet, 10,000 people responded, and after carefully evaluating the responses, we excluded 2683 respondents due to inappropriate answers or lack of time data. Furthermore, to ensure the validity and reliability of the self-reported information, we reconfirmed the abnormal value of the medical checkup together with the time needed for the response at the time of data screening. We also excluded 2787 respondents whose medical checkup results were defective. Finally, 5530 participants were selected for this study. The study was conducted in accordance with the guidelines proposed in the Declaration of Helsinki. The Ethics Committee of the National Institute of Occupational Safety and Health, Japan reviewed and approved the study protocol (H2921). All participants provided web-based informed consent.

### 2.2. Background Characteristics

The participants were surveyed with a self-completed online questionnaire. The questionnaire included sex, age, type of employment (regular staff, temporary worker, contract employee, entrusted employee, other), type of industry (16 categories), and presence or absence of shift work. Lifestyle variables were also self-reported, and included smoking status (current smoker, ex-smoker, non-smoker), alcohol consumption status (non-consumption, once or twice per week, three to five times per week, every day), and engagement in habitual exercise, defined by the Ministry of Health, Labor and Welfare as continual exercise for at least 30 min per day, 2 days per week, over a year or more.

### 2.3. Assessment of Sitting Time

The worker’s living activity-time questionnaire (WLAQ) includes questions pertaining to when individuals perform certain activities such as going to bed, waking, leaving the house, and arriving at and leaving the workplace. With these questions, it is possible to calculate working hours, leisure time, and sleep time. Once we learn the proportional time a participant spends sitting, we can calculate the actual number of minutes per day participants spend sitting or walking/standing during the typical periods in a worker’s life: a) working time, b) leisure time on a workday, and c) holidays. The individual’s average total ST was calculated from ST on workdays and holidays: 5× for weekday ST and 2× for weekend ST are divided by 7. The WLAQ has been shown to have acceptable reliability and validity [21].

### 2.4. Metabolic Syndrome Components

Height, weight, and results of MetS components (abdominal circumference, AC; systolic blood pressure, SBP; diastolic blood pressure, DBP; high-density lipoprotein cholesterol, HDLC; triglycerides, TG; fasting plasma glucose, FPG) were self-reported from medical checkup information within one year. Body mass index (BMI) was computed from self-reported height and weight. MetS was defined according to the Third Report of the Expert Panel on the Detection, Evaluation, and Treatment of High Blood Cholesterol in Adults (Adult Treatment Panel) [22] as meeting three or more of the following criteria: (1) AC ≥102 cm for men and ≥88 cm for women; (2) TG level of ≥150 mg/dL; (3) HDLC level <40 mg/dL for men and <50 mg/dL for women; (4) FPG ≥100 mg/dL, or use of glucose-lowering medications (insulin or oral agents); or (5) SBP ≥130 mmHg and DBP ≥85 mmHg, or receiving treatment for hypertension.

### 2.5. Statistical Analyses

Continuous data are expressed as mean ± standard deviation (SD) and percentages for categorical variables. The variable normality tests were conducted by a normal quantile–quantile plot and the Shapiro–Wilk test. The Student’s unpaired *t*-tests and chi-squared tests for categorical data were used to compare the differences between the sexes. The total sample was automatically categorized into short, middle, and long tertiles for ST for total (≤8.9 h, 9.0 h–11.9 h, ≥12.0 h) and for each domain: occupational (≤5.2 h, 5.3 h–8.1 h, ≥8.2 h), leisure time on workdays (≤2.6 h, 2.7 h–4.1 h, ≥4.2 h), and leisure time on holidays (≤8.0 h, 8.1 h–11.5 h, ≥11.6 h), respectively. Differences in MetS components between categories of each domain-specific ST analysis using the Jonckheere–Terpstra test were adapted to perform tests for trends. We then conducted multiple logistic regression analyses to odds ratios (ORs) and 95% confidence intervals (CIs) to estimate the association of ST and habitual exercise as defined by the Ministry of Health, Labor, and Welfare [23] with MetS and its components. Also, we evaluated the detailed analysis for each domain-specific and total ST with habitual exercise. Multiple logistic regression analyses were performed based on six categories: engagement in habitual exercise or no habitual exercise, combined with each of the short, middle, and long groups of each ST. The short ST with habitual-exercise group was used as the reference to examine their combined association with MetS. For all regression analyses, we used age, sex, smoking, alcohol, and shift work as the confounders. All statistical analyses were performed using SAS, version 9.4 (SAS Institute Japan, Tokyo, Japan), and results were considered significant at *p* < 0.05.

## 3. Results

The general characteristics of the participants, including MetS components and medical status, are shown in Table 1. The mean age of 5530 participants was 44.4 ± 11.1 years, and 432 (7.8%) were included as having MetS. As shown in Table 1, there were significant differences by sex in almost every characteristic, except age.

The differences of MetS components in each ST tertile and habitual exercise are summarized in Table 2. Although working adults with longer total ST, occupational ST, and leisure-time ST on holidays had significantly higher values for most MetS components (*p* for trend < 0.001), the only components that showed a statistically significant association with leisure-time ST on workday were weight and DBP. Also, adults in the no-habitual-exercise group had a significantly lower HDLC and higher TG.

The ORs for MetS with domain-specific ST and habitual exercise are shown in Figure 1 and Appendix A, respectively. When compared to the short ST group (reference), significant ORs for MetS in the long ST group were observed only in leisure-time ST on holidays (OR = 1.43, 95% CI 1.12–1.83). The no-habitual-exercise group had significantly higher ORs of MetS risk (OR = 1.44, 95% CI 1.47–1.80) compared to the habitual-exercise group. The results of the logistic regression analyses examining the risk of MetS for combined habitual exercise and tertiles of each ST (i.e., total, occupational, leisure-time on workdays and holidays) are depicted in Figure 2 and Appendix A. Compared with the reference group (short ST with habitual exercise), the ORs combined with no habitual exercise for MetS were found to be 1.64 (95% CI 1.12–2.39) in total ST and 1.84 (95% CI 1.30–2.59) in leisure-time ST on holidays (OR = 1.84, 95% CI 1.30–2.59), which were significantly higher than the ORs for other subgroups. In contrast, no combination effect for MetS was found for occupational ST (OR = 1.15, 95% CI 0.77–1.70) and leisure-time ST on workdays (OR = 1.22, 95% CI 0.84–1.78) combined with no habitual exercise.

## 4. Discussion

This cross-sectional analysis of working adults showed that the relationships between domain-specific ST and MetS varied depending on how ST occurred; more specifically, only leisure-time ST on holidays showed a clear independent association with an increasing risk of MetS. We also demonstrated that no habitual exercise clearly increased the risk of MetS. What is more, we found that when the long ST group (daily total and on holiday) was combined with no habitual exercise, the risk of MetS was significantly higher than in the combination of the short ST group and habitual-exercise group. However, these significant combination relationships were not shown with occupational ST and leisure-time ST on workdays.

We examined simultaneously the independent and combined associations of domain-specific ST and exercise habits with MetS, obtaining similar results in both associations. One of our main findings, that leisure-time ST on holidays tended to have a detrimental association with MetS, is consistent with the results of previous studies. Several studies [8,9] have reported stronger deleterious associations for screen time (i.e., TV viewing, using a computer, other small screen recreation) as leisure-time ST with MetS. Also, a few studies using accelerometers have shown that the accumulation of leisure-time ST is related to MetS components [24,25]. In addition, the present study shows that the associations with MetS were not found in occupational ST and leisure-time ST on workdays. This is in accord with the findings of previous studies, which reported that occupational ST may have less deleterious effects on cardiometabolic health [26], cardiovascular disease, and diabetes [27] than TV viewing in leisure-time ST. Although data regarding occupational ST and MetS risk are still limited and findings have been mixed regarding the relationships between occupational ST and indicators of MetS health [28,29], our study which included not only TV viewing, but also natural ST on holidays, observed that occupational ST tended to be less adversely associated with MetS. With this in mind, these findings provide evidence that the specific domain of ST itself may be important, not just the amount of ST, and it would be informative if future studies could observe apparent differences in domain-specific ST.

For the identified discrepancy between occupational ST and leisure-time ST on holidays, we do not have the detailed data that would be required to establish why the domain variations in MetS occurred, but we can consider several possibilities. First, it may be that occupational ST has more accompanying breaks (i.e., the breaks theory) than leisure-time ST on holidays. Healy et al. [30] first examined the associations between breaks in objectively assessed ST and MetS components, and they suggest that breaking up ST may provide beneficial metabolic effects. Second, some previous research has suggested that TV viewing and other screen-based entertainment in leisure time may be more associated with unhealthy dietary behavior such as snacking [31,32,33]. Third, occupational ST may be differentially influenced by multi-level sociodemographic and lifestyle factors (e.g., sex, age, household income, and education) [34,35] and occupational characteristics (i.e., job position, occupational PA, and shift work) [36,37]. Generally, the workday of most adults could have a bidirectional relationship with daily activities, and these influences are more complex to interpret compared with time on holiday. Thus, future SB studies assessing the association between ST during workdays (including occupational ST and leisure-time ST on workdays) and health risks need to be conducted.

On the other hand, our results have provided new insights into how habitual exercise is related to domain-specific ST and MetS. The results of our study showed that the no-habitual-exercise group had significantly higher ORs for MetS (OR = 1.44, 95% CI 1.48–1.80) than the habitual-exercise group, after adjusting for all covariates. These results were consistent with those of renowned studies [17,38,39,40] which showed that MetS was inversely associated with participation in PA. This doubtless suggests that habitual exercise can mitigate the detrimental association between ST and MetS. Our results also support those of a previous study [41], which reported that meeting PA recommendations such as 150 min/week of moderate to vigorous PA could mitigate the deleterious association between ST and MetS. Although our study did not quantify exercise, it showed the possibility that a continual exercise habit of at least 30 min per day, 2 days per week, over a year or more may also mitigate the risk for MetS in workers who spend a large amount of time sitting every day. However, further studies are required on why the mitigating effect of habitual exercise was limited to total ST and leisure-time ST on holiday. Holtermann et al. [42] reported that occupational PA increased health risk, while leisure-time PA decreased health risk. These contrasting health effects, called the “PA health paradox” may be complexly related to occupational ST and leisure-time ST on workdays, and even exercise habits. Further study focused on workdays is needed to clarify these relationship.

This study had several strengths and limitations. The main strengths include the information on distinctive SB. That is, leisure time was not limited to TV viewing. Another strength of this study was that we used a validated questionnaire that provided continuous domain-specific ST outcome data; thus, more predictive information was retained than when categorical variables only were considered. At the same time, our study included a large population of workers and encompassed a wide range of employment sectors. Thus, we were able to adjust for occupational characteristics. Some limitations of this study need to be noted, one of which is that the large body of cross-sectional data was collected through the internet, running the risk of questionable legitimacy if secure channels were not used. In addition, ST was calculated based on self-report, although objective measurement devices such as accelerometers can avoid the vagaries of the self-report method. Furthermore, it was not possible for us to determine causality, because this study only carried out a cross-sectional examination.

## 5. Conclusions

Our study confirmed that accumulating daily ST, particularly in leisure time on holiday, can increase the risk of MetS and its components, and that occupational ST may be less harmful than total and leisure-time ST on holiday for MetS risk. This suggests that focusing on a single domain of ST or considering only total ST may blur the distinction in the association between ST and MetS in working adults. This study also provided evidence that habitual exercise may mitigate the MetS risk caused by cumulative specific-domain ST, as well as total ST. Future studies should investigate this theme prospectively to provide causal evidence for a recommendation to keep low amounts of total ST to lessen the odds of developing the MetS. Furthermore, based on our findings, health promotion interventions could reduce the sedentary time on holidays and increase the time spent on exercise.

## Figures and Tables

**Figure 1 ijerph-17-03883-f001:**
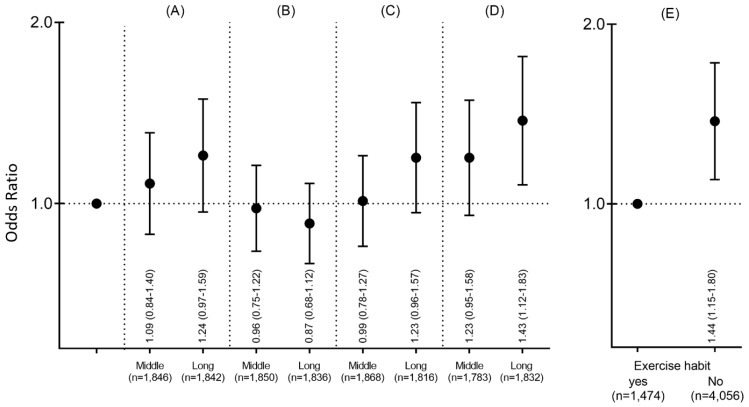
ORs for MetS and (**A**) total ST, (**B**) occupational ST, (**C**) leisure-time ST on workdays, (**D**) leisure-time ST on holidays, and (**E**) habitual exercise. Analysis adjusted for age, sex, smoking (0: ex-smoker and non-smoker, 1: smoker), alcohol (0: non-consumption, 1: once or twice per week, 3–5 times per week, and ≥6 times per week), and shift work (0: absence of shift work, 1: presence of shift work.

**Figure 2 ijerph-17-03883-f002:**
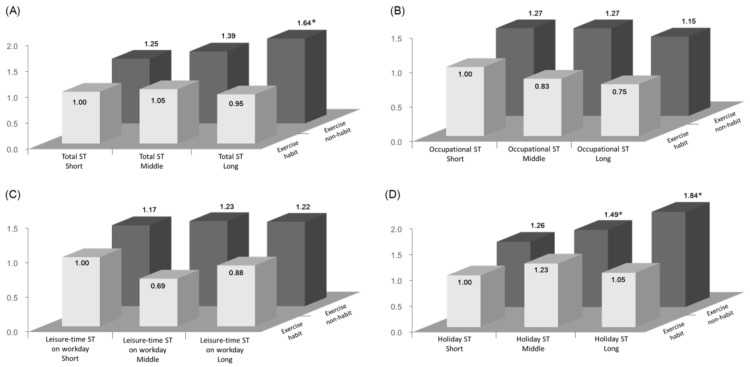
The full-adjusted ORs of MetS cross-classified by level of each ST category: (**A**) total ST, (**B**) occupational ST, (**C**) leisure-time ST on workdays, and (**D**) leisure-time ST on holidays) and habitual exercise among 5530 workers. Analysis adjusted for age, sex, smoking (0: ex-smoker and non-smoker, 1: smoker), alcohol (0: non-consumption, 1: once or twice per week, 3–5 times per week, and ≥6 times per week), and shift work (0: absence of shift work, 1: presence of shift work. * Significant difference (*p* < 0.05) from reference group.

**Table 1 ijerph-17-03883-t001:** Characteristic of the participants.

	Total (*n* = 5530)	Male (*n* = 3868)	Female (*n* =1662)
Age, year	44.4 ± 11.1	44.5 ± 11.0	44.1 ± 11.3
ody weight, kg	64.1 ± 13.1	68.8 ± 11.2	53.1 ± 10.2 *
BMI	22.8 ± 3.8	23.5 ± 3.5	21.1 ± 3.9 *
AC, cm	80.9 ± 10.5	83.1 ± 9.7	74.9 ± 10.2 *
SBP, mmHg	119 ± 15.2	122 ± 14.2	113 ± 15.4 *
DBP, mmHg	76.8 ± 12.6	79.0 ± 12.2	71.6 ± 12.1 *
HDLC, mg/dL	65.1 ± 22.1	62.3 ± 21.5	71.5 ± 22.0 *
TG, mg/dL	110 ± 81.9	119 ± 90.1	88.6 ± 52.9 *
FPG, mg/dL	93.1 ± 22.4	95.0 ± 24.2	88.6 ± 16.9 *
WLAQ			
Working time, h	10.2±1.7	10.5±1.7	9.4±0.8 *
ST during the working time, h	6.3±3.3	6.8±3.2	5.2±3.3 *
ST during the leisuretime on workday, h	3.7±2.0	3.5±1.9	4.1±2.2 *
ST on non-workday, h	9.6±4.1	9.7±4.2	9.2±4.0 *
Medical status, *n* (%)			
MetS	432 (7.8)	349 (9.0)	83 (5.0) *
Hypertension	559 (10.1)	465 (12.0)	144 (8.7) *
Diabetes	208 (3.8)	189 (4.9)	50 (3.0) *
Hyperlipidemia	315 (5.7)	297 (7.7)	97 (5.8) *
Current smoker, *n* (%)	1543 (27.9)	1217 (31.5)	326 (19.6) *
Alcohol consumers, *n* (%)	1892 (34.2)	1501 (38.8)	391 (23.5) *

Values are presented as *n* (%) or mean ± standard deviation. Abbreviations: BMI, body mass index; AC, Abdominal circumference; SBP, systolic blood pressure; DBP, diastolic blood pressure; HDLC, high-density lipoprotein cholesterol; TG, triglycerides; FPG, fasting plasma glucose; MetS, metabolic syndrome; WLAQ, Worker’s Living Activity-time; ST, sitting time. * Significant differences between male and female by Student’s unpaired t-tests (*p* < 0.05).

**Table 2 ijerph-17-03883-t002:** Difference of tertile groups in metabolic syndrome components (*n* = 5530).

	Weight, kg	AC, cm	SBP, mmHg	DBP, mmHg	HDLC, mg/dL	TG, mg/dL	FPG, mg/dL	MetS	Hypertension	Diabetes	Hyperlipidemia
Total ST											
short (≤ 8.9 h, *n* = 1842)	61.5 ± 12.5	79.1 ± 9.9	118.1 ± 15.3	75.4 ± 12.2	67.2 ± 22.9	102.5 ± 71.7	91.6 ± 22.0	123 (6.68)	194 (10.5)	79 (4.29)	131 (7.11)
middle (9.0 h–11.9 h, *n* = 1846)	64.9 ± 13.0	81.3 ± 10.4	120.0 ± 15.0	77.3 ± 12.7	64.7 ± 22.5	111.3 ± 85.2	93.4 ± 23.2	143 (7.75)	183 (9.91)	77 (4.17)	124 (6.72)
Long (12.0 h≤, *n* =1842)	65.9 ± 13.4	82.3 ± 10.8	120.3 ± 15.3	77.7 ± 12.7	63.3 ± 20.7	114.8 ± 87.1	94.1 ± 21.9	166 (9.01)	232 (12.6)	83 (4.51)	139 (7.55)
*P* for trend	<0.001	<0.001	<0.001	<0.001	<0.001	<0.001	<0.001	<0.001	0.045	0.745	0.608
Occupational ST											
short (≤ 5.2 h, *n* = 1844)	61.7 ± 12.9	79.4 ± 10.4	118.9 ± 15.4	75.8 ± 12.6	66.7 ± 22.9	105.0 ± 81.4	91.6 ± 21.9	142 (7.70)	199 (10.8)	86 (4.66)	135 (73.2)
middle (5.3 h–8.1 h, *n* = 1850)	64.2 ± 13.3	81.3 ± 10.3	119.3 ± 15.4	76.6 ± 12.7	65.4 ± 22.5	112.0 ± 82.0	93.3 ± 21.7	148 (8.00)	204 (11.0)	69 (3.73)	128 (6.92)
long (8.2 h≤, *n* =1836)	66.4 ± 12.6	82.1 ± 10.6	120.2 ± 14.9	78.0 ± 12.4	63.1 ± 20.8	111.7 ± 81.9	94.2 ± 23.4	142 (7.73)	206 (11.2)	84 (4.58)	131 (7.14)
*P* for trend	<0.001	<0.001	<0.001	<0.001	<0.001	<0.001	<0.001	0.969	0.678	0.892	0.825
Leisure-time ST on workday											
short (≤2.6 h, *n* = 1846)	64.6 ± 12.3	80.6 ± 9.9	1192 ± 15.1	77.2 ± 12.7	64.8 ± 22.4	108.8 ± 74.6	93.6 ± 24.5	139 (7.53)	193 (10.5)	84 (4.55)	128 (6.93)
middle (2.7 h–4.1 h, *n* = 1868)	64.6 ± 13.2	81.3 ± 10.4	119.6 ± 15.3	76.8 ± 12.5	65.0 ± 21.6	109.6 ± 87.9	92.4 ± 21.0	138 (7.39)	223 (11.9)	85 (4.55)	141 (7.55)
long (4.2 h ≤, *n* =1816)	63.1 ± 13.6	80.9 ± 11.1	119.5 ± 15.3	76.3 ± 12.6	65.4 ± 22.3	110.4 ± 82.3	93.2 ± 21.6	155 (8.54)	193 (10.6)	70 (3.85)	125 (6.88)
*P* for trend	<0.001	0.711	0.617	0.032	0.441	0.874	0.942	0.259	0.858	0.302	0.957
Leisure-time ST on holiday											
short (≤8.0 h, *n* = 1915)	63.0 ± 12.4	79.9 ± 10.0	118.3 ± 15.3	76.1 ± 12.5	65.8 ± 22.8	105.2 ± 71.7	92.3 ± 22.9	121 (6.32)	188 (9.82)	79 (4.13)	125 (6.53)
middle (8.1 h–11.5 h, *n* = 1783)	64.1 ± 13.4	80.9 ± 10.5	119.4 ± 15.0	77.1 ± 12.4	65.0 ± 22.0	109.9 ± 88.7	92.4 ± 21.9	137 (7.68)	193 (10.8)	76 (4.26)	130 (7.29)
long (11.6 h ≤, *n* =1832)	65.3 ± 13.5	82.1 ± 10.8	120.7 ± 15.4	77.2 ± 12.9	64.3 ± 21.5	113.7 ± 84.5	94.5 ± 22.2	174 (9.50)	228 (12.5)	84 (4.59)	139 (7.59)
*P* for trend	<0.001	<0.001	<0.001	<0.001	0.089	0.010	0.005	<0.001	0.010	0.490	0.205
Habitual-exercise											
Yes (*n* = 1804)	64.5 ± 12.4	80.8 ± 10.1	119.6 ± 14.6	76.6 ± 12.0	67.0 ± 22.6	105.9 ± 76.5	93.1 ± 22.3	116 (6.43)	185 (10.3)	70 (3.88)	136 (7.54)
No (*n* = 3726)	63.9 ± 13.4	81.0 ± 10.7	119.4 ± 15.6	76.9 ± 12.9	64.1 ± 21.9	111.4 ± 84.3	93.0 ± 22.4	316 (8.48)	424 (11.4)	169 (4.54)	258 (6.92)
*P* for difference	0.113	0.688	0.661	0.517	<0.001	0.025	0.945	<0.001	0.211	0.261	0.405

## Data Availability

Data is not deposited in publicly available repositories due ethical restrictions and participant confidentiality concerns. However, on reasonable request, derived data supporting the findings of this study are available with approval from principal investigator (Dr. Tomoaki Matsuo) and the Research Ethics Committee of this study.

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
