# Peer review of "The Effect of Domain-Specific Sitting Time and Exercise Habits on Metabolic Syndrome in Japanese Workers: A Cross-Sectional Study"

_ijerph, 2020, doi:10.3390/ijerph17113883_

Round 1

Reviewer 1 Report

Thank you for the opportunity to review your manuscript titled The effect of domain-specific sitting time and exercise habit on metabolic syndrome in Japanese workers: A cross-sectional study.The study is well prepared and relevant to detect the risk of sitting time and exercise habit on metabolic syndrome. I have a couple of comments for the authors:

In the abstract, the authors write the acronyms ST and MetS without specifying their meaning the first time they appear. Please indicate their meaning the first time they appear.

Please use MeSH terms (https://www.ncbi.nlm.nih.gov/mesh/) for keywords. Sedentary behavior is the only one they have in this format. Using these types of words will help you search for your manuscript in databases.

Participants and procedures: In the text they indicate that the survey started in 2018, but they don't indicate the specific period of time that the interview lasted, accessible to the participants. Can you specify what that time period was?

Author Response

Responses to the reviewers

The authors would like to thank the reviewers for their constructive critique to improve the manuscript. We made every effort to address the issues raised and to respond to all comments. The answers are indicated in red font in the manuscript. Please, find next a detailed, point-by-point response to the reviewers' comments.

Responses to comments from Reviewer #1:

Comment #1

In the abstract, the authors write the acronyms ST and MetS without specifying their meaning the first time they appear. Please indicate their meaning the first time they appear.

Response #1:

We would like to thank the reviewer for the comment. We have revised the abbreviation according to the reviewer’s suggestion. The revised part in the Abstract is as follows: “The effects of domain-specific (i.e., occupational, leisure-time on workday, and holiday) sitting time (ST), and exercise on Metabolic syndrome (MetS) development are insufficient”.

Comment #2

Please use MeSH terms (https://www.ncbi.nlm.nih.gov/mesh/) for keywords. Sedentary behavior is the only one they have in this format. Using these types of words will help you search for your manuscript in databases.

Response #2:

We would like to thank the reviewer for the constructive comment. We have revised the keywords based on the MeSH terms and provided the following:

sedentary behavior, occupational health, leisure activity, sitting, physical activity

Comment #3

Participants and procedures: In the text they indicate that the survey started in 2018, but they don't indicate the specific period of time that the interview lasted, accessible to the participants. Can you specify what that time period was?

Response #3:

Following the reviewer’s suggestion, we have added a description for survey period in the Participants and procedures section on page 5, line 13, as follows:

“This cross-sectional, internet-based survey was conducted from January to July 2018 with participants answering a self-completed questionnaire.”

Reviewer 2 Report

Abstract well described the study.

Introduction may be too brief. Suggest to provide more extensive lit review which can further support the research question. 

Methods were clearly described. 

Describe the normality assumption used. 

Were those with other chronic diseases eg. hypertension etc were excluded in Table 2? Number of samples for this table should be stated in title. I don't think the footnote actually described the analysis presented here. 

Figure 1 & 2 - Results from regression analysis best presented in table. 

Suggest implications on practice and recommendations for future research in this area. 

Author Response

Responses to the reviewers

The authors would like to thank the reviewers for their constructive critique to improve the manuscript. We made every effort to address the issues raised and to respond to all comments. The answers are indicated in red font in the manuscript. Please, find next a detailed, point-by-point response to the reviewers' comments.

 Responses to comments from Reviewer #2:

 Comment #1                  

Introduction may be too brief. Suggest to provide more extensive lit review which can further support the research question.

Response #1:

We would like to thank the reviewer for the comment. We agree with his/her opinion. Following his/her suggestion, we have enhanced the Introduction and provided more references in this section of the revised manuscript.

Specifically, we have added a new part in the Introduction on page 4 lines 10–18 as follows:

“However, most of these studies focused on fragments of daily activity, such as watching television (TV) or using computers in leisure time. Nevertheless, sitting time (ST) occurs generally across numerous domains on a daily basis [10]. To fully understand the ST role as a factor related to health risks, it is necessary to identify whether various unequal domain-specific ST (i.e., occupational and leisure-time ST on workdays and holidays) have a similar effect on MetS. In addition, there is evidence, which suggested that the adverse health effects of different domain ST combined may be greater than those associated with each ST separately [11]. Therefore, it is important to consider the unequal contribution of each specific-domain ST and the total amount of ST.” 

Comment #2                  

Describe the normality assumption used.

Response #2:

Following the reviewer’s suggestion, we have added a description of normality assumption in the Statistical analyses section.

Please refer to page 7 lines 21–22 as follows:

“The variable normality tests were conducted by a normal quantile-quantile plot and the Shapiro-Wilk test.”

Comment #3                  

Were those with other chronic diseases eg. hypertension etc were excluded in Table 2? Number of samples for this table should be stated in title. I don't think the footnote actually described the analysis presented here.

Response #3:

We would like to thank the reviewer for pointing this out. We apologize for this mistake. Please note that we have added the analysis results for chronic diseases (MetS, hypertension, diabetes, and hyperlipidemia). Moreover, we have checked the entire Table 2 and revised it after considering your comment.

Please refer to Table 2.

 Comment #4                  

Figure 1 & 2 - Results from regression analysis best presented in table.

Response #4:

We would like to thank the reviewer for his/her comments. We agree with his/her comment that a Table was needed to understand the logistic regression analysis. Therefore, we have added the supplement Tables related to Figures 1 and 2. In addition, revised Figures to avoid duplication of results (delete of ORs). We think that showing the results in Figures and supplement Tables would help the reader's understanding.

Please refer to Tables S1, S2 and the Results section.

Comment #5                  

Suggest implications on practice and recommendations for future research in this area.

Response #5:

Following the reviewer’s suggestion, we have added a description in the Conclusion section regarding the future research on page 12 line 24–page 13, line 2 of the revised manuscript, as follows:

“Future studies should investigate this theme prospectively to provide causal evidence for a recommendation to keep low amounts of total ST to lessen the odds of developing the MetS. Furthermore, based on our findings, health promotion interventions could reduce the sedentary time in holidays and increase the time spent with exercise.”

Reviewer 3 Report

The authors analyzed effects of domain-specific sitting time and exercise habits on metablol ic syndrome in large cohort of 5530 Japanese workers.

The results of the study are  original and of high practical value for large group of experts dealing with the problem of deleterious effects of sitting time and protective role of habitual exercise in large context of public health, cardiometabolic disordes and occupational medicine.

In my opinion the authors should better justify that self-reported data from study participants devoted to METS components from recent medical checkup information ( i.e height, weight, abdominal circumference , blood pressure, TG, HDLC etc ) are enough reliable.

I was not able to find among the 39 references the paper of Holtermann et al (43) quoted in discussion.

In conclusion I recommend to publish the manuscript after minor correction/supplement information.

Author Response

Responses to the reviewers

The authors would like to thank the reviewers for their constructive critique to improve the manuscript. We made every effort to address the issues raised and to respond to all comments. The answers are indicated in red font in the manuscript. Please, find next a detailed, point-by-point response to the reviewers' comments.

Responses to comments from Reviewer #3:

Comment #1

In my opinion the authors should better justify that self-reported data from study participants devoted to METS components from recent medical checkup information ( i.e height, weight, abdominal circumference , blood pressure, TG, HDLC etc ) are enough reliable.

Response #1:

We appreciate the reviewer’s comment. The data collection through the Internet have a risk of questionable legitimacy. As suggested, we have added the necessary information in the Participants and procedures section on page 5, line 21–page 6 line 2 of the revised manuscript as follows:“To ensure the reliability of the response, we asked to prepare medical checkup information within 1 year before start this survey, and when they were not prepared, they could not proceeded to the main survey. Through the Internet, 10,000 people responded, and after carefully evaluating the responses, we excluded 2,683 respondents due to inappropriate answers or lack of time data. Furthermore, to ensure the validity and reliability of self-reported information, we reconfirmed the abnormal value of the medical checkup together with the time needed for the response, at the time of data screening.”

Comment #2

I was not able to find among the 39 references the paper of Holtermann et al (43) quoted in discussion.

Response #2:

We apologize to the reviewer for this mistake. We have corrected the reference number in the revised manuscript.

Round 2

Reviewer 2 Report

Thank you to accepting the suggestions and improving the manuscript based on the feedback given.